# A selective role for ventromedial subthalamic nucleus in inhibitory control

**Benjamin Pasquereau[1,2,3,4], Robert S Turner[1,3,4]***

[1]Department of Neurobiology, University of Pittsburgh, Pittsburgh, United States; [2]Institut des Sciences Cognitives Marc Jeannerod, CNRS UMR 5229, Bron, France; [3]Center for Neuroscience, University of Pittsburgh, Pittsburgh, United States; [4]Center for the Neural Basis of Cognition, University of Pittsburgh, Pittsburgh, United States

**Abstract** The subthalamic nucleus (STN) is hypothesized to play a central role in the rapid stopping of movement in reaction to a stop signal. Single-unit recording evidence for such a role is sparse, however, and it remains uncertain how that role relates to the disparate functions described for anatomic subdivisions of the STN. Here we address that gap in knowledge using non-human primates and a task that distinguishes reactive and proactive action inhibition, switching and skeletomotor functions. We found that specific subsets of STN neurons have activity consistent with causal roles in reactive action stopping or switching. Importantly, these neurons were strictly segregated to a ventromedial region of STN. Neurons in other subdivisions encoded task dimensions such as movement *per se* and proactive control. We propose that the involvement of STN in reactive control is restricted to its ventromedial portion, further implicating this STN subdivision in impulse control disorders.

DOI: https://doi.org/10.7554/eLife.31627.001

## Introduction

Although considerable effort has been directed to identify the neural circuits involved in action initiation, much less attention has been given to neural substrates of response suppression. The ability to inhibit inappropriate action is an important component of executive functions necessary for adaptive behavior. Because deficient forms of response inhibition are associated with many impulse control disorders (*Robbins et al., 2012*; *Bari and Robbins, 2013*), a better understanding of the inhibition brain-network should help improve treatments for those patients. One of the most common paradigms for investigating inhibitory control is the stop-signal task (*Aron, 2011*; *Schall and Godlove, 2012*; *Verbruggen and Logan, 2008*). Subjects are required to initiate a movement in response to a Go signal, but in a small fraction of trials, a subsequent Stop signal instructs subjects to withhold the impending movement. This kind of reactive stopping becomes increasingly more difficult as the time interval between Go and Stop signals lengthens. Stop-signal task performance is well explained by a race model in which the Go and Stop signals trigger independent processes that compete to be the first to finish (*Boucher et al., 2007*; *Logan et al., 1984*; *Verbruggen and Logan, 2009*). Recent studies have suggested that reactive stopping behavior is actually composed of a cascade of movement-suppressive processes that depend on transmission of activity along multiple brain pathways (*Pouget et al., 2017*; *Schmidt and Berke, 2017*).

Neuroimaging and lesion studies have posited a critical role for the subthalamic nucleus (STN) in stopping actions (*Aron and Poldrack, 2006*; *Aron et al., 2007*; *Li et al., 2008*; *Sharp et al., 2010*). Because the STN stands at the crossroad between 'hyperdirect' and indirect basal ganglia (BG) pathways, this structure occupies an unique position from which it is hypothesized to block Go-related activity conveyed by the BG direct pathway (*Mink, 1996*; *Nambu, 2004*; *Nambu et al., 2002*).

**\*For correspondence:**
rturner@pitt.edu

**Competing interests:** The authors declare that no competing interests exist.

Specifically, STN lesions result in impulsive and perseverative behaviors (*Baunez et al., 1995*; *Wiener et al., 2008*), with a reduced ability to suppress action in the stop-signal task (*Eagle et al., 2008*; *Obeso et al., 2014*). In parkinsonian patients, deep brain stimulation (DBS) of STN has a variety of effects on the stopping of pre-planned or on-going actions (*Mirabella et al., 2012*; *Obeso et al., 2013*; *Ray et al., 2009*; *Swann et al., 2011*; *van den Wildenberg et al., 2006*), and beta frequency local field potentials are modulated by inhibitory processes (*Alegre et al., 2013*; *Bastin et al., 2014*; *Benis et al., 2014*; *Kühn et al., 2004*; *Ray et al., 2012*).

Although the STN is commonly modeled as a key node in action suppression (*Nambu et al., 2002*; *Frank, 2006*), direct electrophysiologic evidence remains weak. Recent single-unit recordings from rats showed that success or failure on the stop-signal task correlated with the timing of STN responses evoked by the stop-signal (*Schmidt et al., 2013*). The picture is far from complete, however. These short-latency STN responses, for example, were not selective to the Stop cue (*Mallet et al., 2016*), but were also present for Go cues that prompted a switch in behavior (i.e. the selection and initiation of an alternate action). This illustrates the fact that paradigms used to study action suppression rarely control for potentially confounding factors such as the attentional detection of the stop cue (*Duann et al., 2009*; *Erika-Florence et al., 2014*; *Hampshire et al., 2010*) or behavioral switching (*Crone et al., 2006*; *Isoda and Hikosaka, 2007*; *Isoda and Hikosaka, 2008*; *Rushworth et al., 2002*). Consequently, the degree to which STN neuronal activity is involved specifically in response inhibition remains unclear. Indeed, one non-human primate study reported that a sub-population of STN neurons exhibit a change of activity when an animal is required to switch from an automatic (i.e., pre-potent) action to one guided by sensory instructions (*Isoda and Hikosaka, 2008*). Such a mechanism could account for many of the previous observations.

In addition, it has been unclear how these hypothesized roles for STN in action inhibition relate to the known organization of STN into anatomically and functionally distinct territories (*Nambu et al., 2002*; *Alexander et al., 1990*; *Parent and Hazrati, 1995*; *Hamani et al., 2004*). The posterior-dorsal-lateral STN is interconnected with a basal ganglia circuit devoted to skeletomotor function, whereas associative- and limbic-related territories are found in progressively more anterior, ventral and medial regions of the STN (*Haynes and Haber, 2013*; *Mettler and Stern, 1962*; *Monakow et al., 1978*; *Shink et al., 1996*; *Wichmann et al., 1994*). We predicted that activity correlated with movement inhibition would be concentrated in the anterior ventromedial regions of the STN known to receive direct afferents (*Inase et al., 1999*; *Künzle, 1978*) from stop-associated regions of prefrontal cortex (*Aron et al., 2004*).

To determine if the STN conveys signals consistent with its predicted role in action suppression, we trained two monkeys to perform a novel stop-signal task that disentangled processes that are often confounded in studies of inhibitory control. We hypothesized that a subset of STN neurons exhibits a short-latency change in activity consistent with a selective role in stopping. Here we tested that hypothesis by studying single unit activity in the STN while monkeys performed action inhibition and switching tasks. Our results support an active role for the STN in stopping, and refine current theories by implicating a discrete anterior ventromedial region of the STN in inhibitory control, further confirming that this site is critical for impulse control disorders (*Baup et al., 2008*; *Mallet et al., 2008*).

## Results

### Behavioral suppression of on-going action

Two monkeys (C and H) were trained to perform a Go/NoGo-countermanding arm movement task (*Figure 1*) that combines two paradigms commonly used to investigate the neuropsychology of inhibitory functions: the stop-signal task and the Go/NoGo task. Specifically, early in a trial, during a start position hold period, an animal was instructed how to respond to an upcoming 'trigger' stimulus: in 50% of trials a Go cue was presented (green stimulus; referred to hereafter as *go* trials) instructing animals to respond to the trigger stimulus by initiating a targeted reaching movement; in the other 50% of trials a NoGo cue (red stimulus; referred to as *no-go* trials) instructed animals to withhold a response to the trigger and continue holding the hand at the start position. On a minority of trials (25% each of Go and NoGo conditions), the trigger stimulus was followed by a switch-signal (blue stimulus) instructing the animal to countermand the already-triggered nascent response. If the

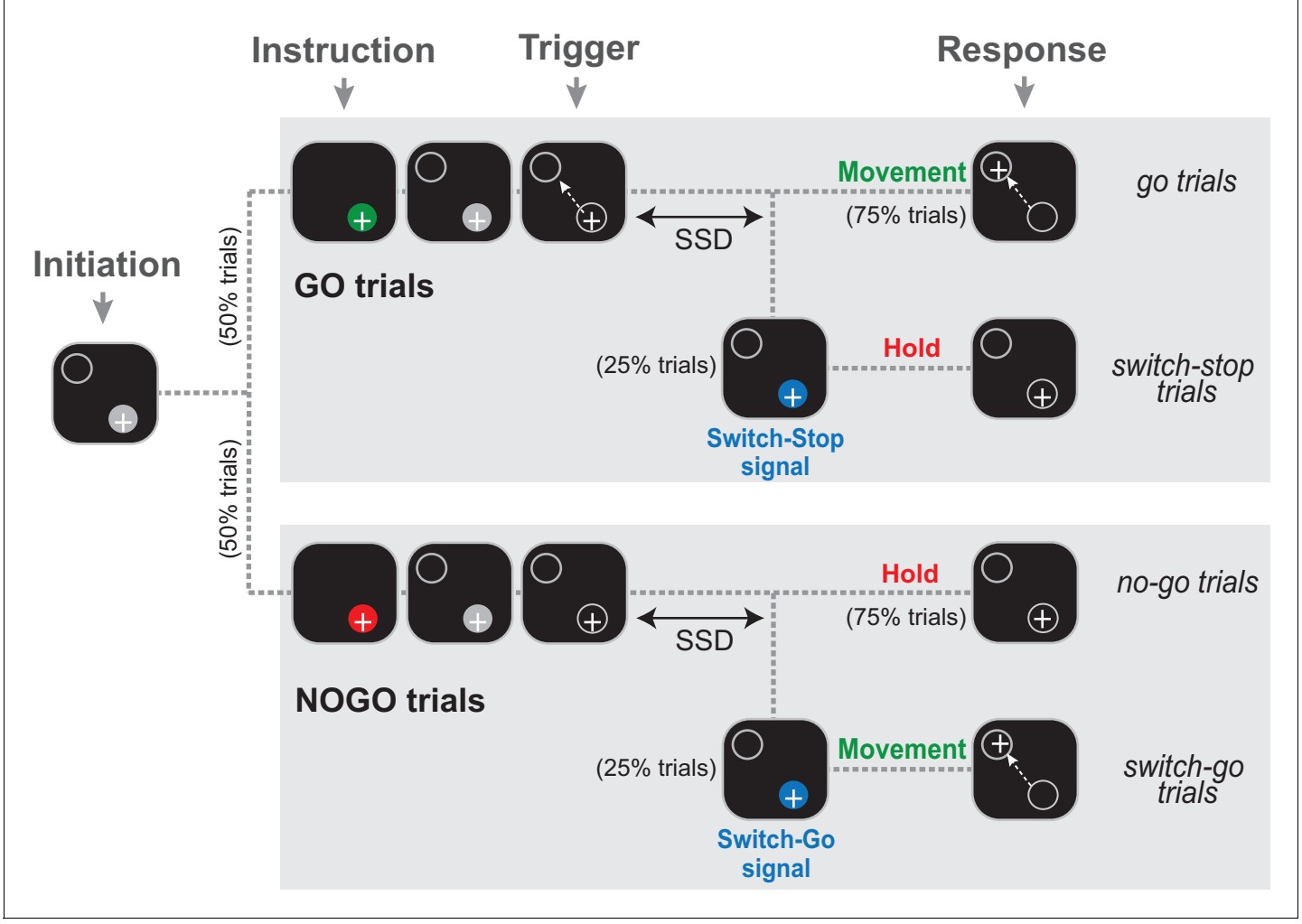

**Figure 1.** The Go/NoGo-countermanding task. Temporal sequence of visual displays for the four trial types. After the animal initiated a trial by moving the hand-controlled cursor (+) to the start position (gray circle), one of two possible instruction cues (selected at random between trials) were presented briefly. A green cue indicated a Go condition, while a red cue indicated a NoGo condition. Depending the condition, the animal was then required to respond to the Trigger stimulus by moving to the target (*go trials*) or continuing to hold the cursor in the start position (*no-go trials*). On a small fraction of trials (25%), the Trigger stimulus was followed after a variable delay (the switch-signal delay, SSD) by a switch-signal (blue cue) that instructed the animal to countermand the initially planned response.

DOI: https://doi.org/10.7554/eLife.31627.002

switch-signal occurred during a Go trial, the animal had to suddenly suppress the triggered movement and hold the initial position (referred as *switch-stop* trials). In these trials, the switch-signal was equivalent to a stop-signal in a traditional stop-signal task. Alternatively, if the switch-signal occurred during a NoGo trial, the animal was required to rapidly initiate a reaching movement (referred to as *switch-go* trials). Thus, the same blue switch-signal had different meanings (stop vs. go) depending on which type of trial it was used in. The target for reaching movements was presented at one and the same location for all trial types and a food reward was delivered at the end of each successful trial.

As is typical for stop-signal tasks (*Scangos and Stuphorn, 2010*; *Mirabella et al., 2011*), median RTs for *switch-stop* trials in which the monkeys failed to suppress the response (referred as *stop-failure* trials) were faster than median RTs detected for *go* trials (Mann-Whitney U-test, p<0.001). *Figure 2* shows that the RT distributions calculated across sessions for *stop-failure* trials (C: median = 555 ± 64 ms; H: median = 502 ± 77 ms) were shifted to the left with respect to the RT distributions of *go* trials (C: median = 580 ± 58 ms; H: median = 516 ± 93 ms). Thus, consistent with the

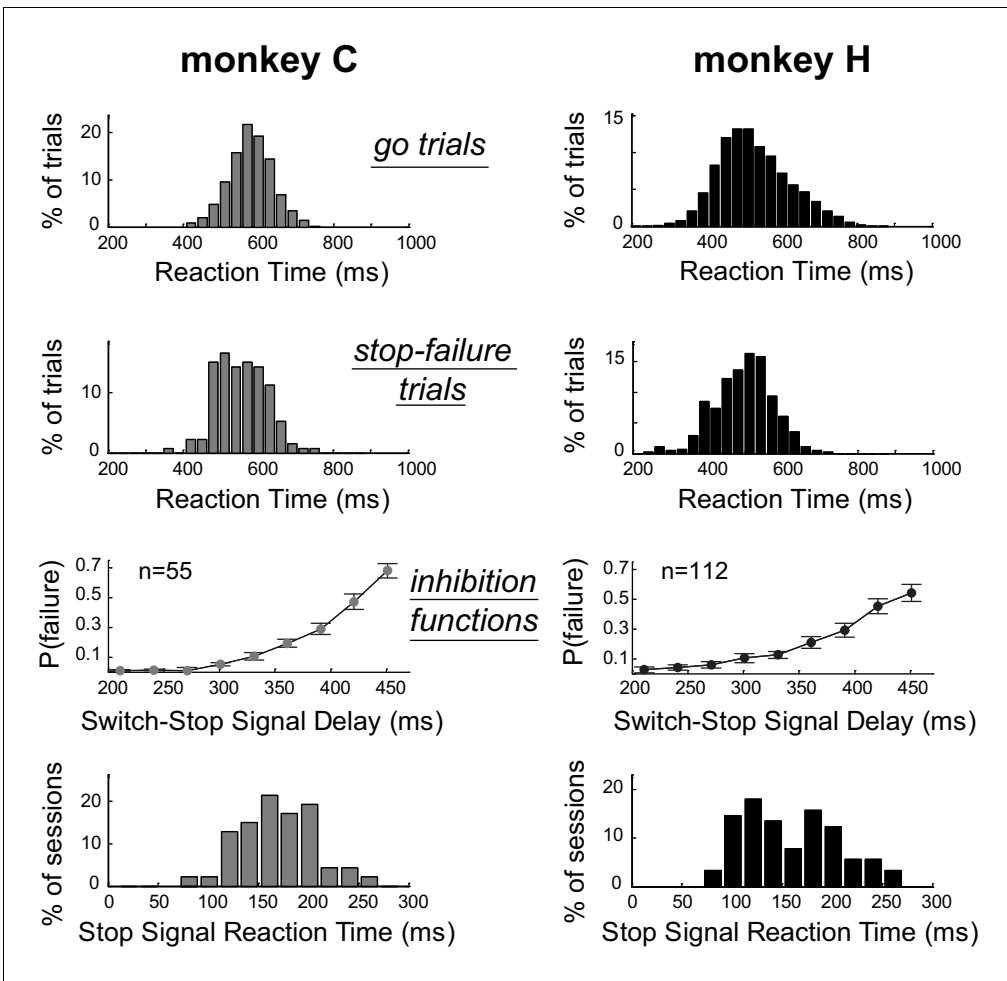

**Figure 2.** Estimation of the time required to inhibit the planned response. Subject performance in our version of the stop-signal task. The reaction time distributions for *go* trials and *stop-failure* trials were calculated across recording sessions. *Stop-failure* trials corresponded to those on which the animal failed to inhibit the planned response. Inhibition functions averaged across sessions were obtained by plotting the proportion of *stop-failure* trials as a function of the switch-signal delay (mean ±SEM). By combining inhibition functions with the reaction time distributions, we estimated the stop-signal reaction times (SSRT) for each session. The bottom row of plots show the distributions of SSRT estimates for each monkey across recording days.

DOI: https://doi.org/10.7554/eLife.31627.003

race model (*Boucher et al., 2007*; *Logan et al., 1984*), reaching movements that monkeys could not stop despite the presentation of switch-signal were largely those that had the shortest RTs. As shown in *Figure 2*, the incidence of *stop-failure* trials increased (inhibition functions: linear regressions, p<0.05) as a function of the delay between the go trigger signal and the switch-signal (i.e., the *switch-signal delay*, SSD). In other words, the longer the SSD, the more difficult it became for an animal to stop the nascent triggered movement. This kind of behavior is commonly modeled by a race model in which Stop and Go processes compete to be the first to finish (*Logan et al., 1984*). For each individual behavioral session, we used a race model to estimate the stop-signal reaction time (SSRT, see Materials and Methods), which is an estimate of the maximum delay within which a triggered response can be stopped successfully (*Congdon et al., 2012*; *Band et al., 2003*). Distributions of SSRTs estimated session by session from each monkeys' behavior are shown in *Figure 2*. We calculated that the average SSRT during our task was around 160 ms (C: median = 169 ± 37 ms; H: median = 152 ± 47 ms). The SSRTs had similar distributions for the two animals (Mann-Whitney U-test, p=0.14), but were slightly longer than those reported in earlier studies that used a manual stop-signal task [~140 ms (*Scangos and Stuphorn, 2010*);~150 ms (*Mirabella et al., 2011*)]. The

difference is likely attributable to higher-level cognitive aspects of our novel paradigm, in which monkeys had to integrate different task events to identify the appropriate behavioural response (i.e., stopping or going).

## STN activities related to switch-stop signal

While the monkeys performed the Go/NoGo-countermanding task, we recorded single-unit activity from 167 neurons in the right STN (55 from monkey C; 112 from monkey H). To determine whether and how the STN was involved in action suppression, we investigated Stop-related activity in this sample of neurons by comparing the firing rate during successful *switch-stop* trials with the firing rate recorded during those *go* trials in which the response would have been stopped if the switch-signal had been presented at the equivalent SSDs (referred to as latency-matched trials, see Materials and methods). *Figure 3A–B* show the activity, aligned to switch-stop signal, for two example STN neurons. We compared activity between switch-stop and go conditions using a sliding window procedure combined with receiver operating characteristic (ROC) analysis. The area under the ROC curve (AUC) reflected the degree to which firing rates differed reliably between conditions. The logic of our task set two criteria a single neuron's activity was required to meet to qualify as being involved in the stop process. First, the activity must differ between trials in which a response is initiated versus when it is cancelled. Second, this difference (i.e. the neural cancellation time) must occur within the SSRT; otherwise, it would be too late to mediate response inhibition (*Hanes et al., 1998*; *Paré and Hanes, 2003*). Following this line of reasoning, the discharge rates for both example STN neurons (*Figure 3A–B*) diverged significantly after switch-stop signal presentation (2-tailed *t*-test, p<0.01). The evolution of AUC values over time indicated that the Stop-related changes in activity occurred within the SSRT, with a neural cancellation time preceding the behavioral response inhibition (-48 ms and -34 ms in *Figure 3A and B*, respectively). Inspection revealed distinct response patterns in the two example neurons, however. The activity shown in *Figure 3A* exhibited a short-latency phasic increase in discharge during switch-stop trials at times suitable for an active role in stopping. We tested for this type of 'switch-stop' activity as a significant decrease of AUC values (p<0.01) during the SSRT (*blue shaded* area of AUC curve, *Figure 3A*). Other STN neurons showed a pre-movement buildup of activity during *go* trials and that buildup was truncated abruptly during the SSRT of *switch-stop* trials (e.g., *red shaded* area of AUC curve, *Figure 3B*). Neurons with this type of activity were termed 'Movement' cells because the pattern of activity approximated what might be observed in motor output structures such as primary motor cortex. Movement cells were characterized by a significance increase in AUC values (p<0.01). Within the race model framework, the switch-stop pattern was consistent with an active participation in the stop process whereas the movement-like pattern more likely reflected involvement in successful termination of the competing go process.

Of the 167 neurons recorded, 63 (38%) qualified as being involved in the stop process (see table in *Supplementary file 1*). *Figure 3C* illustrates AUC values for each qualifying neuron. The average neural cancellation time was −55 ms before the SSRT (C: median = −44 ± 32 ms; H: median = −58.5 ± 25 ms). A large majority (78%; 49 of 63) of these STN neurons were classified as Movement cells. For both monkeys, the population-averaged activity illustrated in *Figure 4A* confirmed that preceding successful stopping, the activity of Movement cells initially resembled that seen during *go* trials consistent with a buildup of movement-related activity. However, after appearance of the switch-stop signal, that buildup was terminated precipitously and the population firing rate decreased below the baseline discharge rate. The rapidity of this decrease in firing suggested more the involvement of an active mechanism in stopping than a simple deceleration in the Go process. Importantly, a second subset (22%; 14 of 63) of STN neurons responded to the switch-stop cue with a transient increase in firing rate suitable for an active role in the stop process. These Switch-stop cells (*Figure 4A*) not only distinguished when a response had to be canceled, but they did so with a transient increase of activity that began early enough to contribute directly to the behavior (median = −32 ± 25 ms). The mean neural cancellation times were slightly earlier for Movement cells than for switch-stop cells (median = −60 ± 48 ms; Mann-Whitney U-test, p=0.042) making it unlikely that the truncated buildup in movement cells was a product of switch-stop activity.

Remarkably, Switch-stop cells were found in only the most anterior, ventral, and medial portion of the STN (*Figure 4B*). The same anatomic segregation was observed in each animal considered independently. This anatomic 'hotspot' is a territory well-described as receiving inputs from associative

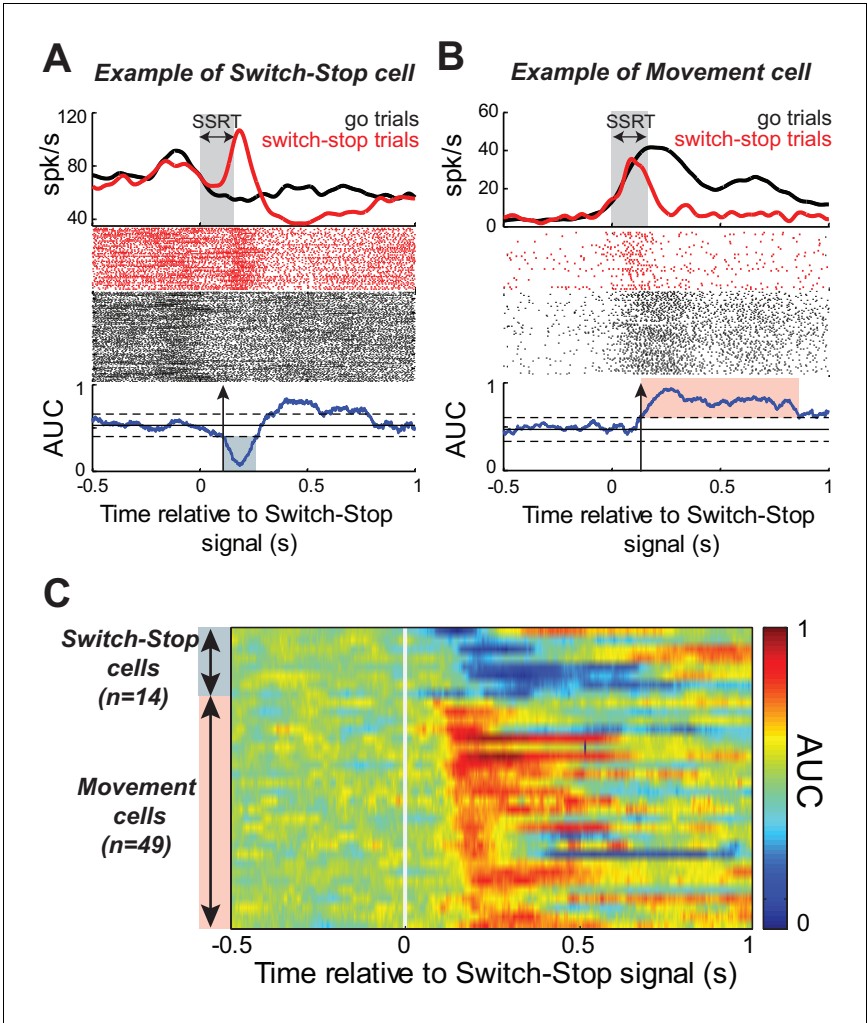

**Figure 3.** Response of STN neurons to switch-stop signal. The activity of two exemplar neurons that were classified as a (A) Switch-stop cell and (B) Movement cell. Spike density functions and rasters were constructed around the presentation of the switch-stop signal (*red*) and latency-matched *go* trials (*black*). In the spike density functions, the gray shading indicates the stop-signal reaction time (SSRT) estimated from concurrent behavior. A sliding window ROC analysis compared firing rates between successful *switch-stop* trials and latency-matched *go* trials. *Blue lines* (below rasters): areas under the ROC curve (AUC) reflecting statistical difference between spike count distributions for each time step (60 ms window-width with a step of 1 ms). Horizontal lines: baseline ± threshold for significant AUC (2-tailed *t*-test, p<0.01). Vertical arrow: detected neural cancellation time. Only a subset of trials is shown in the raster of *go* trials to aid visualization of spike occurrences. (C) Color plot of all AUC values for neurons in which the neural cancellation time occurred within the SSRT. Colors reflect the relative direction and intensity of the stop-related response (blue = stop related relative increase in firing; red = stop related relative decrease in firing; AUC scale in the bar).

DOI: https://doi.org/10.7554/eLife.31627.004

and limbic cortical areas (*Haynes and Haber, 2013*; *Karachi et al., 2005*) and as implicated in impulse control disorders (*Baup et al., 2008*; *Mallet et al., 2008*). It is interesting to note that neuronal activities related to reward delivery or the occurrence of errors in task performance did not show a similar localization within the STN (see *Figure 4—figure supplement 1*). No differences were found in the mean baseline firing rates of the different subtypes of STN neurons or between the two animals (2-way ANOVA, $F < 2.13$ p>0.15).

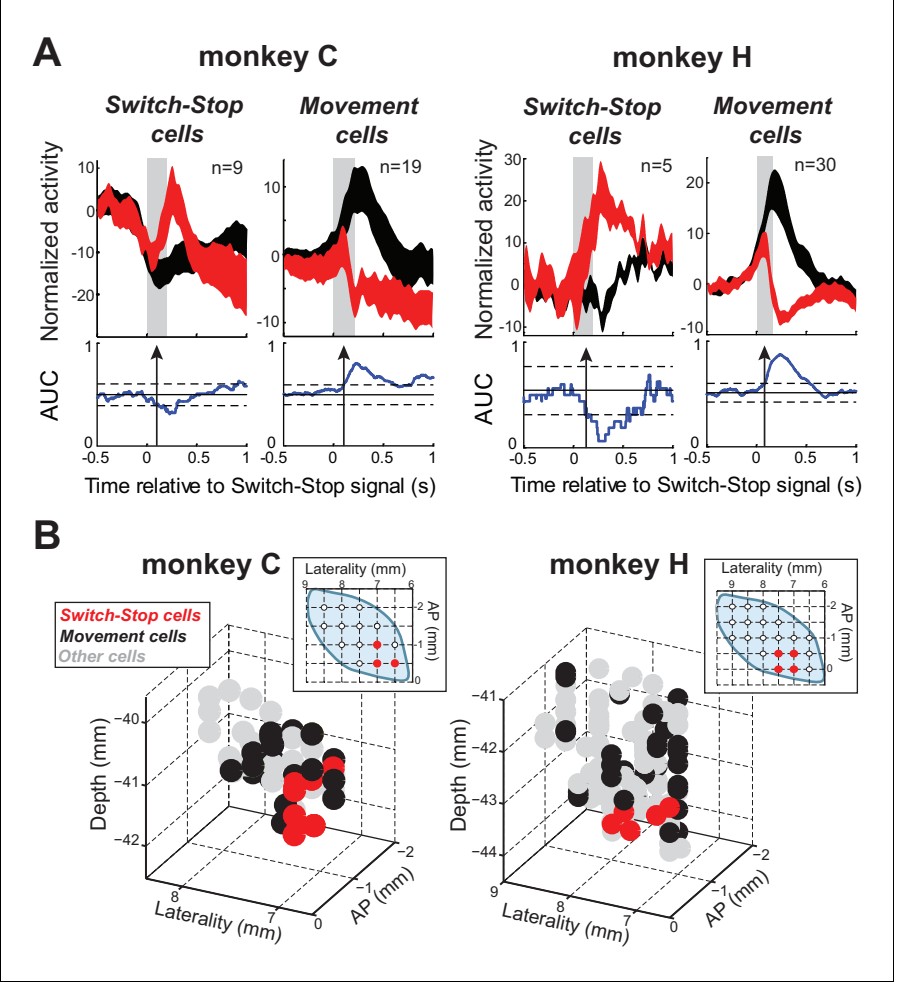

**Figure 4.** Two types of STN response during stopping. (**A**) Population-averaged activities of STN neurons that showed a neural cancellation time within the SSRT. Spike density functions aligned on switch-stop signal presentation (*red*) and the equivalent time in latency-matched go trials (*black*) were normalized by subtracting the baseline activity (500 ms before the signal) and grouped according to the response pattern evoked in neuronal activity during stopping: increase or decrease in firing relative to latency-matched go trials (*Switch-stop* and *Movement* cells, respectively). The width of the spike density function line indicates the population SEM. Otherwise, these figures follow the conventions of *Figure 3*. (**B**) Topography of cell types in the STN. Two- and three-dimensional plots of cell type distributions based on coordinates from the recording chamber. AP: anterior-posterior plane.

DOI: https://doi.org/10.7554/eLife.31627.005

The following figure supplement is available for figure 4:

**Figure supplement 1.** Response of STN neurons to error and successful trials.

DOI: https://doi.org/10.7554/eLife.31627.006

## Comparison of cancelation time for STN neurons and muscles

We recorded EMGs from four different arm muscles (biceps longus, triceps lateralis, anterior deltoid and posterior deltoid) while monkey C performed the behavioral task. The inhibition of arm movement was accomplished by both a rapid relaxation of agonist muscles and a brief activation of antagonists (*Figure 5A*). To test whether Switch-stop and Movement cells recorded in monkey C were suitable for driving such muscle patterns, we compared the median cancelation times between those STN neurons and muscle activities (*Figure 5B*). Note that this comparison controls for any inaccuracies in estimation of the SSRT, since the same estimation was used for both neuronal and EMG activities. Although we found that the cancelation times of Switch-stop and Movement cells preceded

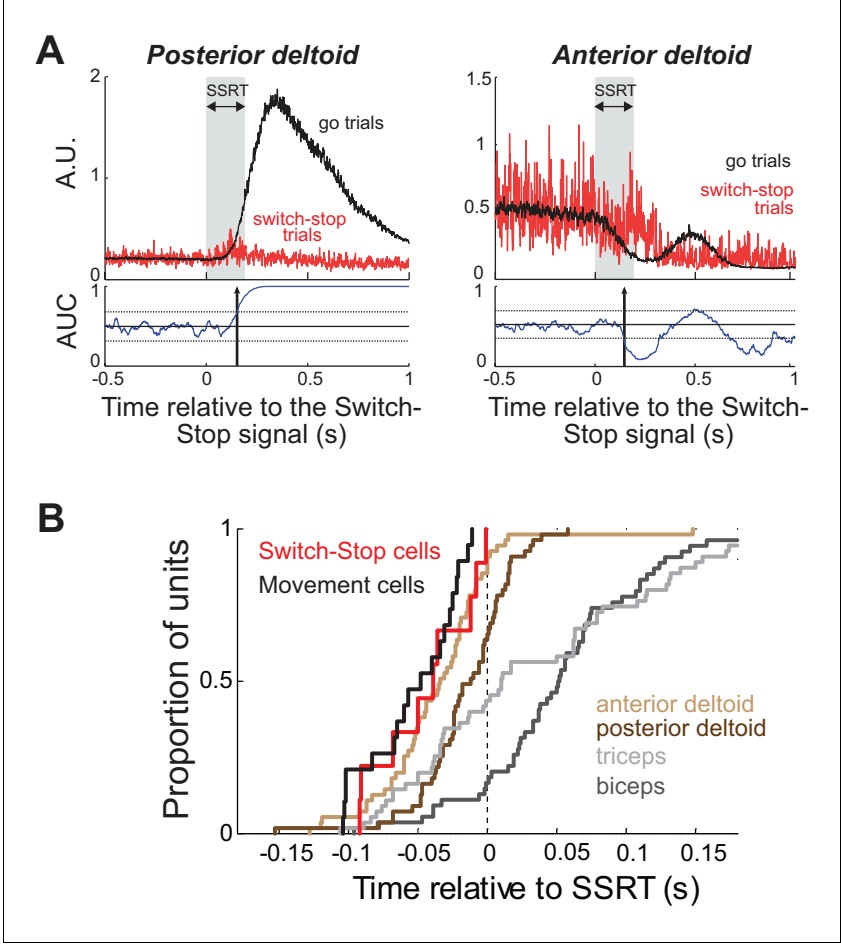

**Figure 5.** Muscle and neural cancelation times in monkey C. (**A**) EMG signals collected from agonist (*left*) and antagonist (*right*) muscles were aligned on the presentation of the switch-stop signal and latency-matched go trials. Vertical arrow: detected muscle cancellation time. (**B**) Cumulative proportion of STN neurons (Switch-stop vs. Movement cells) and muscles with cancelation times relative to the SSRT boundary. The average cancelation time of the anterior deltoid (−34 ms before SSRT) preceded the other EMG activities in the stop process (Mann-Whitney U-test, p<0.05).

DOI: https://doi.org/10.7554/eLife.31627.007

the majority of stop-related EMG activities (posterior deltoid = −30 ms, triceps = −51 ms, biceps = −94 ms; Mann-Whitney U-test, p<0.05), no significant time lag was measured between STN signals and the changes of activity in the anterior deltoid (−10 ms; Mann-Whitney U-test, p>0.05). Thus, in order play a causal role in the earliest changes in muscle activity related to action suppression; it would be necessary for STN to transmit stop-related activity via an extremely rapid pathway.

## STN activities related to stopping or switching

Does the activity of switch-stop neurons reflect a specific signal related to rapid stopping of a nascent motor command? Or is switch-stop activity more closely related to attentional processing of the visual cue (*Duann et al., 2009*; *Erika-Florence et al., 2014*; *Hampshire et al., 2010*) or to behavioral switching (*Isoda and Hikosaka, 2008*)? We addressed this question by comparing a neuron's switch-stop activity against the activity observed in the same neuron around the time when the animal was asked unexpectedly to initiate a reaching movement (i.e., during *switch-go* trials). The population-averaged response of Switch-stop cells differed considerably between *switch-stop* trials (*Figure 6A*) and *switch-go* trials (*Figure 6B*) and there was a good deal of between-neuron variability in response profiles during *switch-go* trials (compare confidence intervals of the population

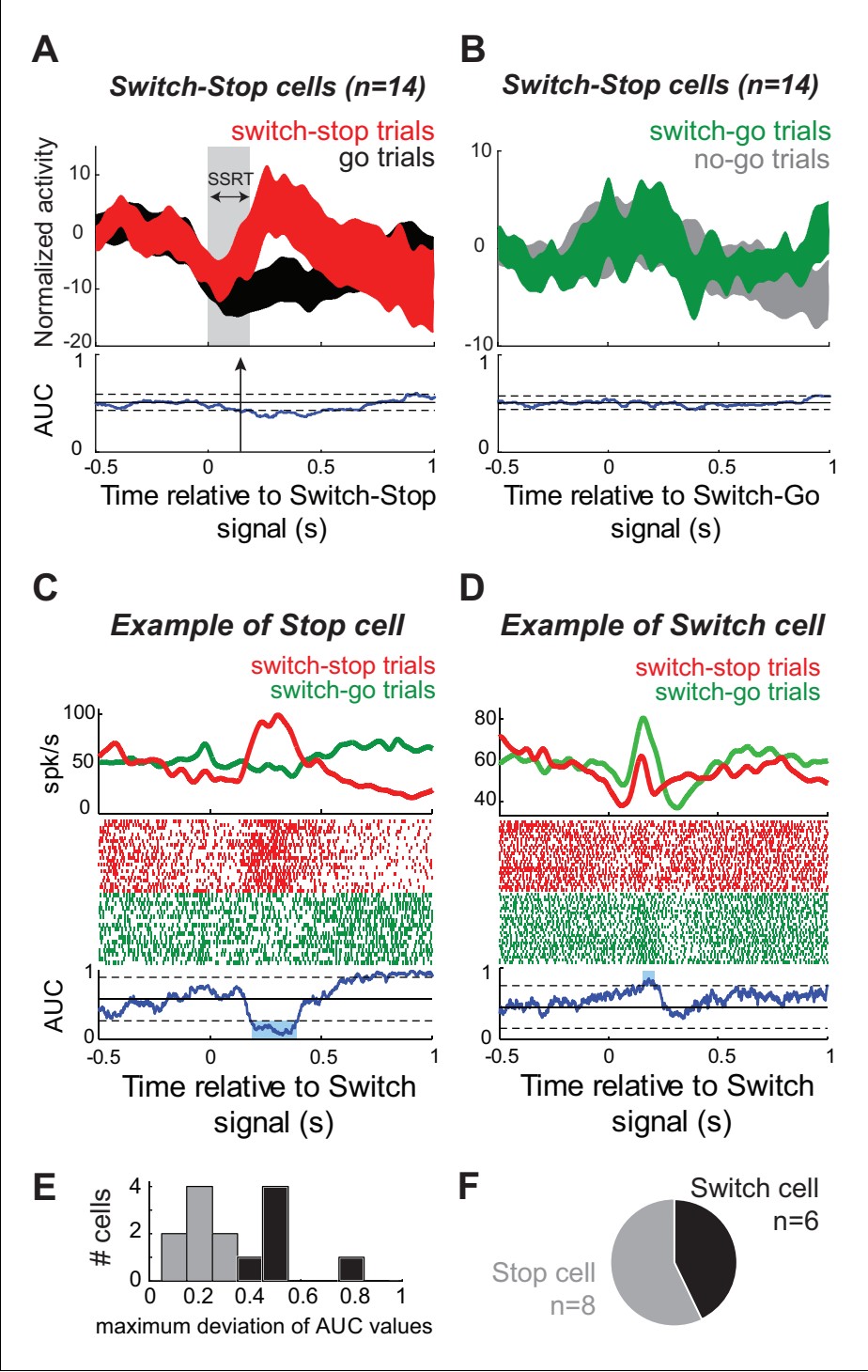

**Figure 6.** A subset of STN neurons encode a selective stop signal. (A–B) Population-averaged activities of STN neurons that showed short-latency increases of discharge in response to the switch-stop signal presentation. Spike density functions (mean ±SEM) were aligned on (A) the switch-stop signal or (B) the switch-go signal. These figures follow the conventions of *Figure 3*. (C–D) The activity of two exemplar neurons classified as (C) Stop cell and (D) Switch cell. These spike density functions and rasters were constructed around the presentation of both types of switch signal (switch-stop and switch-go). To compare the firing rate between trials, we used the same ROC analysis (p<0.01). (E–F) The maximum deviation of AUC values was used to categorize STN neurons as Stop cells (AUC values significantly reduced relative to the control period) or Switch cells (AUC values not significantly reduced relative to the control period).

*Figure 6 continued on next page*

*Figure 6 continued*

DOI: https://doi.org/10.7554/eLife.31627.008

The following figure supplement is available for figure 6:

**Figure supplement 1.** STN activities independent to eye position or saccade.

DOI: https://doi.org/10.7554/eLife.31627.009

averages between *Figure 6A–B*). For individual Switch-stop cells, responses were compared between successful *switch-stop* and *switch-go* trials. As shown for an example cell in *Figure 6C*, a neuron was judged to be involved specifically in stopping if its response during *switch-stop* trials was significantly larger than activity during the equivalent period of *switch-go* trials (2-tailed *t*-test, p<0.01). Alternatively, if a neuron's response during *switch-go* trials was roughly equal to (p>0.01) or greater than (p<0.01, e.g., *Figure 6D*) activity during *switch-stop* trials, then that activity was considered more likely to be related to attentional or switching functions. Of the 14 Switch-stop cells, 8 (57%) were classified as Stop cells (*Figure 6E–F*). In the remainder (43%; 6 of 14), here termed Switch cells, responses were not selective for the condition when the cue signaled need for a rapid stop. The activity of Switch cells may be more closely associated with behavioral switching or visual attentional processes. No difference was found in the neural cancellation times between Stop and Switch cells (Mann-Whitney U-test, p=0.66).

Similar phasic signals related to stopping have been observed in frontal eye fields (*Hanes et al., 1998*) and superior colliculus (*Paré and Hanes, 2003*) of monkeys performing an eye movement countermanding task (*Schall and Godlove, 2012*). Because the STN includes an oculomotor territory (*Matsumura et al., 1992*), we investigated whether the activity of Switch-stop cells could be related to eye movements made in the task. Our animals did not move their eyes in a consistent way before or during the presentation of the Switch-stop signal (*Figure 6—figure supplement 1*). Regression analyses found no Switch-stop cells (0 of 14; Permutation test, p>0.01) whose activity encoded eye position or the timing of saccades (*Figure 6—figure supplement 1*). Thus, it is unlikely that Switch-stop activity can be attributed to oculomotor behavior.

## Control for sensorimotor activity

Next, we considered the possibility that apparent Stop or Switch activities in the STN were related to the activation of antagonist muscles during stopping (see *Figure 5*). Activity in the STN that correlates closely with limb movements and, conceivably, activation of individual muscles is concentrated in the skeletomotor territory of the STN (*Wichmann et al., 1994*; *Georgopoulos et al., 1983*; *Patil et al., 2004*; *Theodosopoulos et al., 2003*). Short latency responses to proprioceptive stimuli are also most common in the skeletomotor STN (*Wichmann et al., 1994*; *DeLong et al., 1985*). If Stop or Switch activities were simply a correlate of skeletomotor commands, then Stop and Switch cells would be expected to have other characteristics of skeletomotor neurons such as responsiveness to proprioceptive stimulation. To investigate this possibility, we tested for short-latency (<200 ms) neuronal responses to brief torque motor-induced movements of the monkey's shoulder and elbow joints. *Figure 7A* shows a typical short-latency increase in discharge evoked by a sudden torque impulse (2-tailed *t*-test, p<0.01). Of the 167 neurons, 42 (25%) exhibited short-latency responses (*Figure 7B*). These 'Torque' responsive neurons were concentrated in the dorsal postero-lateral STN (*Figure 7C*), agreeing well with previous descriptions of the skeletomotor territory (*Parent and Hazrati, 1995*; *Haynes and Haber, 2013*). Many Torque cells (38%; 16 of 42) were also Movement cells (see above; *Figure 7D*). In contrast, only one Torque cell was categorized as a Switch cell (2%; 1 of 42; *Figure 7E–F*). Furthermore, Torque cells were concentrated distant from the anterior ventromedial region where Stop and Switch cells were located (*Figure 7C*). Therefore it is unlikely that the Stop and Switch activities described above were related in a direct way to skeletomotor commands such as the activation of antagonist muscles.

## STN activities related to proactive control

Another potentially important distinction is that between reactive control, required in Switch-Stop and Switch-Go tasks, and a type of proactive control used in Go/NoGo-countermanding tasks (*Figure 8*) (*Aron, 2011*; *Stuphorn and Emeric, 2012*). We assessed our task's ability to manipulate

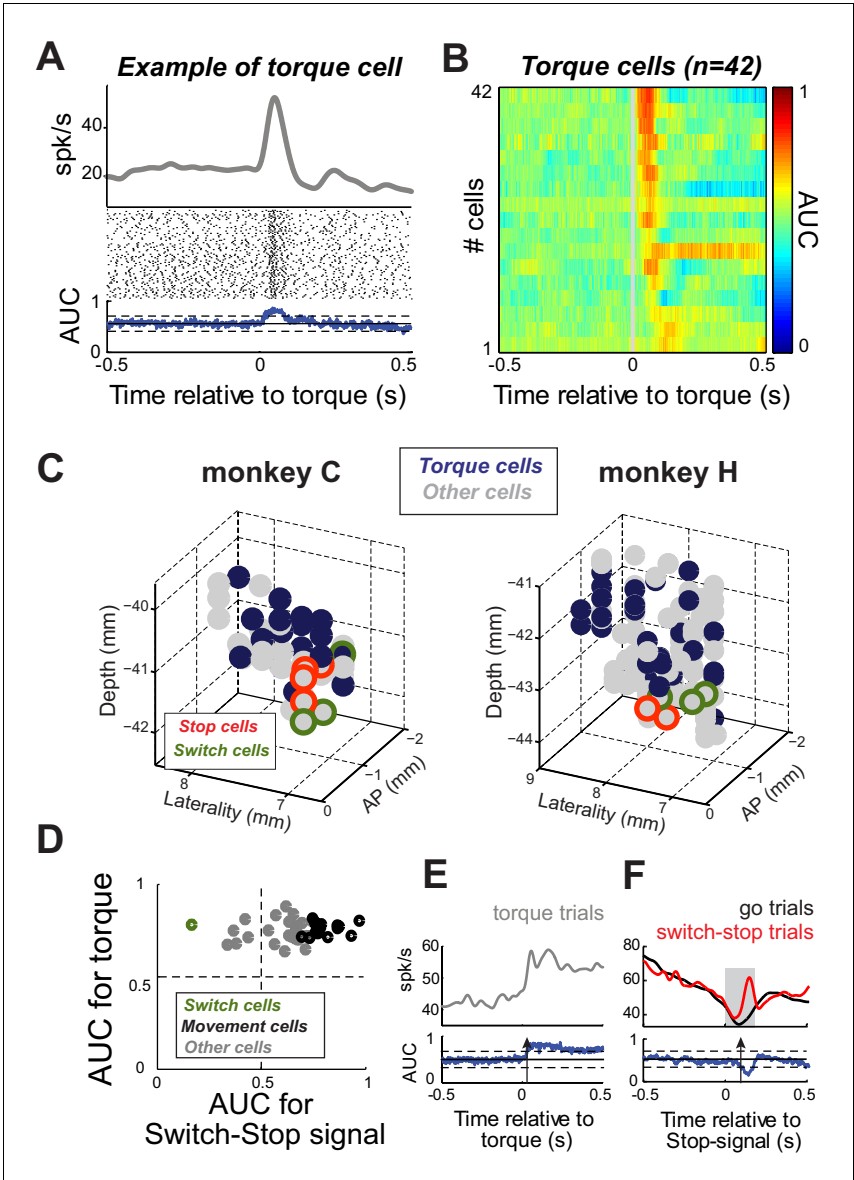

**Figure 7.** Proprioceptive stimulation evokes STN responses. (**A**) Example of STN neuron that exhibited a rapid change of activity in response to torque-induced proprioceptive stimulation (2-tailed *t*-test, p<0.01). (**B**) Color plot of AUC values calculated over time for each neuron responding at short-latency to torque pulses (<200 ms; p<0.01; AUC color scale in the bar). (**C**) Topography of cell types in the STN. Note that torque-responsive neurons were located primarily in the dorso-lateral STN and did not overlap with switch-stop neurons. (**D**) For each neuron with a significant torque response, the peak AUC for the torque response is plotted versus the peak AUC for the effect of stopping. Note that only one torque-responsive neuron had a *switch*-type response (*green* symbol). (**E–F**) For this one Torque cell also categorized as a Switch cell, spike density functions and AUC values are separately aligned on (**E**) torque pulses and (**F**) switch-stop signals. The methods and conventions used to compare muscle or neuronal activities were described in *Figure 3*. The arrows in AUC values indicate the times of first differential activity.

DOI: https://doi.org/10.7554/eLife.31627.010

proactive control by measuring the difference in RTs between *go* trials and *switch-go* trials. Evidence that the NoGo instruction induced a proactive behavioral set to *not respond* included a significant prolongation of RTs in *switch-go* trials relative to those in *go* trials (Mann-Whitney U-test, p<0.001;

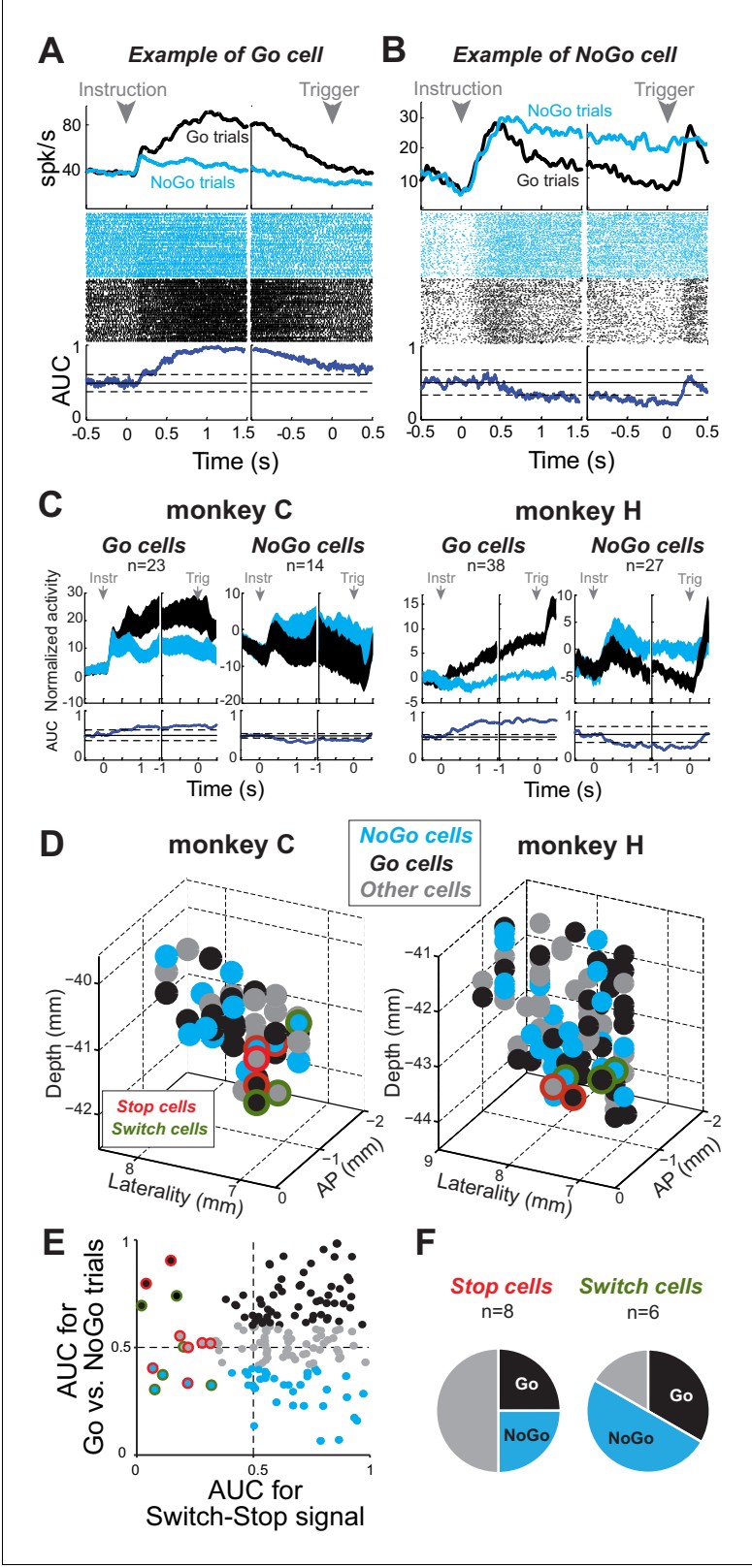

**Figure 8.** Proactive encoding of action suppression in STN. (**A–B**) The activity of two exemplar neurons that were classified as a Go cell (**A**) and a NoGo cell (**B**). Spike density functions, rasters and AUC values were constructed around the presentation of the instruction cue and the time of the response trigger. The categorization was determined according to the AUC values present during the time interval between those task events (thresholds:

*Figure 8 continued on next page*

*Figure 8 continued*

p<0.01). (**C**) Population-averaged activities for STN Go and NoGo neurons (p<0.01). Spike density functions were normalized by subtracting the baseline activity (500 ms before the instruction). (**D**) Topography of cell types in the STN. (**E–F**) Distribution of peak AUC values for the Go/NoGo comparison plotted versus the AUC values for the switch-stop comparison. (**F**) No relationship was observed in the STN between both reactive and proactive encodings ($\chi^2$=1.75, p=0.42).

DOI: https://doi.org/10.7554/eLife.31627.011

The following figure supplements are available for figure 8:

**Figure supplement 1.** Estimation of the preparation cost.
DOI: https://doi.org/10.7554/eLife.31627.012
**Figure supplement 2.** Response of STN neurons to switch-go signal.
DOI: https://doi.org/10.7554/eLife.31627.013

see *preparation cost* in *Figure 8—figure supplement 1*), which did not differ between monkeys (Mann-Whitney U-test, p=0.82).

We then tested for correlates of proactive control in the STN by comparing neuronal firing rate between *go* and *no-go* trials. For some neurons (e.g., *Figure 8A*), firing rates following instruction delivery and the subsequent instructed delay period were markedly higher during *go* trials than *no-go* trials (2-tailed *t*-test, p<0.01). For other neurons, activity during this period was significantly higher during *no-go* trials (e.g., *Figure 8B*). We used AUC values from the instructed delay period to categorize neurons into Go- and NoGo-related categories. Go neurons (37%; 61 of 167) showed elevated tonic firing during the instructed delay period of *go* trials relative to that of *no-go* trials (*Figure 8C*) and thus were considered unlikely to play an important role in proactive impulse control. NoGo cells (25%; 41 of 167) showed elevated delay period activity during *no-go* trials. Neither category showed evidence of anatomic clustering within the STN (*Figure 8D*). We found no obvious interaction between the categories of neurons associated with reactive and proactive control ($\chi^2$ test, $\chi^2$ = 1.75, p=0.42). Roughly equal proportions of Switch-stop cells were categorized as Go cells (4 of 14) and NoGo cells (5 of 14), while the remainder showed no preferential activity during the instructed delay period (*Figure 8E–F*). In parallel, we also found that, of the STN neurons that responded to the Switch-go signal, equal proportions ($\chi^2$=1.14, p=0.29) were categorized as Go cells (31%; 19 of 61) and NoGo cells (29%; 12 of 41), reinforcing the view that reactive and proactive neuronal processes did not interact, even during a disinhibitory phase of the task when an animal had to initiate a movement unexpectedly (*Figure 8—figure supplement 2*). Together, these results suggest that two distinct mechanisms support reactive and proactive modes of action control.

## Discussion

The original idea that the STN is a key node in the response inhibition brain-network arose primarily from evidence from neuroimaging (*Aron and Poldrack, 2006*; *Aron et al., 2007*; *Li et al., 2008*; *Sharp et al., 2010*) and lesion (*Baunez et al., 1995*; *Eagle et al., 2008*; *Obeso et al., 2014*; *Nishioka et al., 2008*) studies. However, many aspects of neuronal signaling during action suppression remained unclear despite recent STN electrophysiological recordings performed in rats (*Schmidt et al., 2013*), monkeys (*Isoda and Hikosaka, 2008*) and humans (*Bastin et al., 2014*; *Benis et al., 2014*; *Ray et al., 2012*; *Benis et al., 2016*). Here, we found that neurons located in the most anterior and ventromedial portion of the STN show a phasic change in activity suitable for an active role in stopping. Our novel paradigm allowed us to confirm that some STN activity is Stop-selective rather than reflecting a variety of potentially conflated factors. To our knowledge, these results provide the first evidence for a selective encoding of stopping at the cellular level in the healthy primate STN.

The stop-signal task has been used widely to investigate the neural substrate of response suppression in oculomotor (*Hanes et al., 1998*; *Paré and Hanes, 2003*; *Ito et al., 2003*; *Stuphorn et al., 2000*) and skeletomotor (*Bastin et al., 2014*; *Scangos and Stuphorn, 2010*; *Mirabella et al., 2011*; *Benis et al., 2016*; *Chen et al., 2010*) systems. This paradigm imposes tight timing constraints in that a neural response must occur within the calculated SSRT for it to contribute to stopping. More than one-third of our STN neurons met this temporal criterion (*Figures 3–*

*4*). Among those cells, the most common response pattern was a movement-related activation that decayed rapidly in response to the stop-signal, similar to the activity observed in agonist muscles (*Figure 5A*). Notably, the second response pattern consisted of a phasic increase in discharge in response to the stop-signal. The co-existence of these two response patterns is consistent with roles for the STN in both the execution of movement and stopping movement (discussed in more detail below). The race model, in its simplest form, predicts that Stop-related activity should precede or at least coincide with the truncated buildup of movement-related activity. We found, however, that the mean neural cancellation time for Movement cells was slightly earlier than that for Stop cells. More-over, Stop-related activity preceded the earliest stop-related truncation of EMG activity by a mere 10 ms. One potential conclusion is that the exact position of the STN in models of reactive stopping may require modification contingent on identification of brain areas with earlier Stop-related activi-ties. An alternative interpretation, that STN Stop-related activity is transmitted to motor centers via a fast-conducting pathway, is discussed below.

The classic stop-signal task, when used by itself as in previous neurophysiologic studies (*Aron and Poldrack, 2006*; *Schmidt et al., 2013*; *Scangos and Stuphorn, 2010*), has the potential to conflate several distinct cognitive processes (*Duann et al., 2009*; *Erika-Florence et al., 2014*; *Hampshire et al., 2010*). First, because the stop-signal is both infrequent (15–30% of trials) and behaviorally relevant, some neural responses to the stop-signal may reflect attentional processing of an unexpected visual cue. Indeed, the STN is part of the network in which visual attention and the salience of events are implemented (*Baunez and Robbins, 1997*; *Baunez et al., 2007*; *Bočková et al., 2011*; *Schmalbach et al., 2014*). Second, stop-signal task performance requires sev-eral aspects of executive control including changes in cognitive set, enhancement of sensory respon-siveness, facilitation of a desired response and performance monitoring. Thus, some phasic responses to a Stop-signal may reflect switching-related sub-processes.

Our stop-signal paradigm disentangled those factors by presenting an identical visual stimulus to signal unexpected commands either to suppress an incipient action or to initiate a reaching move-ment. *Switch-stop and switch-go* trial types were similarly infrequent and behaviorally relevant and both required a behavioral switch from habitual to controlled responding. Comparison of STN activi-ties between the two types of switch trials identified two categories of short-latency responses (*Fig-ure 6*). Although standard criteria would have classified all as playing an active role in stopping, 43% of cells with short-latency stop responses responded similarly or more strongly with sudden action initiation in *switch-go* trials. Quite consistent with the role in attentional or switching processes pro-posed by Isoda and Hikosaka (*Isoda and Hikosaka, 2008*), these STN neurons modified their activity when an animal had to shift behavior rapidly from default to a more attention-demanding controlled behavior.

The second subset of STN neurons were activated selectively only when an animal had to cancel the on-going action on *switch-stop* trials. These cells did not overlap, categorically or anatomically, with the STN neurons most likely to be involved directly in the activation of antagonist muscles, namely those responsive to proprioceptive stimulation and concentrated in the skeletomotor terri-tory of the STN (*Nambu et al., 2002*; *Parent and Hazrati, 1995*; *Baron et al., 2002*). Together, these results provide consistent support for the presence of a selective encoding of reactive stop-ping in a subset of STN neurons after controlling for a range of potential confounds.

Few previous studies have examined STN single-unit activity during performance of a response inhibition-type task. *Schmidt et al. (2013)* described a population of STN neurons in the rat that responded at short-latency to a Stop cue. Notably, unlike the present study, those neurons also responded at roughly equal latency and magnitude to the Go cue, which was presented on all trials (*Mallet et al., 2016*). Finally, Schmidt et al. provided evidence that stop-selective activity appeared in an anatomically-restricted region of the substantia nigra reticulata (SNr), a BG output nucleus located downstream from the STN. Many factors differed between the Schmidt study and ours, including species, tasks, and collection and analysis methods, any combination of which may account for the differences between results. In a recent study in human subjects undergoing DBS surgery for Parkinson's disease, *Benis et al. (2016)* described distinct sub-populations of STN neurons that responded selectively and at short latency to *Stop* and *Go* task signals, similar to our findings. *Isoda and Hikosaka (2008)* recorded from the macaque STN during a task that required animals to switch behaviors between familiar and unexpected oculomotor responses, somewhat similar to our switch-go type trials. Like us, they found a distinct sub-population of STN neurons that responded

selectively and at short latency to the unexpected switch stimulus. No previous study, however, dissociated STN activities related to reactive stopping, reactive switching and proactive (NoGo) response control, tested for potential overlap between stop-related activity and skeletomotor responsiveness, or addressed whether stop-related activity is concentrated in a sub-region of the STN.

Finally, similar movement- and stop-related response patterns have been found in frontal eye cortex and superior colliculus of macaques performing an eye movement countermanding task (*Schall and Godlove, 2012*; *Hanes et al., 1998*; *Paré and Hanes, 2003*). Although the STN includes an oculomotor territory (*Matsumura et al., 1992*), we found no evidence that the Stop-related activity identified here actually reflected movements of the eyes (*Figure 6—figure supplement 1*).

Use of the Go/NoGo task (*Bari and Robbins, 2013*; *Aron, 2011*; *Dalley et al., 2011*) also allowed us to study STN involvement in the sustained ('proactive') maintenance of a movement-suppressive behavioral set between the time of NoGo instruction and the presentation of a trigger stimulus seconds later. Fully a quarter of STN neurons exhibited sustained activity preferentially after a NoGo cue (*Figure 8*). These cells were not confined to a particular territory of the STN, a result that contrasted sharply with the anatomically-restricted location of *stop* cells. The distribution of NoGo cells throughout the STN implies that many functional sub-circuits through the STN are activated during this behavior. It is likely that NoGo activity, as identified here, actually reflects of a variety of independent cognitive and motor processes, analyses of which will require further refinement of behavioral paradigms. Previous studies have implicated the striatum and the striato-pallidal-STN 'indirect' pathway in proactive control (*Casey et al., 1997*; *Kelly et al., 2004*; *Smittenaar et al., 2013*; *Vink et al., 2005*). Although the present results do not address that possibility, they do argue that proactive and reactive movement suppressions involve categorically different neural mechanisms. Not only were *Stop* and NoGo activities distributed differently within the STN, but the presence of NoGo activity in a single-unit had no predictive power over whether that unit also responded preferentially to *switch-stop* or *switch-go* signals (*Figure 8E* and *Figure 8—figure supplement 2*).

The STN receives topographically-organized glutamatergic inputs from most regions of the frontal cortex (*Nambu et al., 2002*; *Haynes and Haber, 2013*; *Monakow et al., 1978*). Among these, the inferior frontal cortex (IFC; potential homologue in macaques: area 45) has been linked closely to reactive stopping based on lesion (*Aron et al., 2003*; *Swick et al., 2008*) and stimulation (*Verbruggen et al., 2010*; *Wessel et al., 2013*) studies. Although the specific topography of the IFC–to–STN projection has yet to be determined, tract-tracing studies indicate that most prefrontal areas [i.e., ventromedial, orbitofrontal, anterior cingulate and dorsal prefrontal cortices (*Haynes and Haber, 2013*)] project to ventral and medial portions of the STN that encompass the regions where *Stop* and *Switch* cells were located. This portion of the STN also receives strong GABAergic inputs from the rostral-medial external pallidum (part of the BG associative circuit) and ventral pallidum (of the BG limbic circuit) (*Haynes and Haber, 2013*; *Shink et al., 1996*; *Karachi et al., 2005*), and additional glutamatergic inputs from the parafascicular nucleus of the thalamus, also a limbic circuit-connected nucleus (*Sadikot et al., 1992*). Consistent with its afferent connectivity, this ventromedial region contained few cells with Movement-like activity (*Figure 4B*), making it unlikely that Stop and Go processes are compared locally within the ventromedial STN.

What efferent pathway allows stop-related activity in the STN to inhibit motor execution at short latency? The present results do not address that question directly, but they do put considerable constraints on the viable answers. A commonly accepted model states that the first efferent step from the STN is a highly divergent projection to the output nuclei of the BG [i.e., internal pallidum (GPi) and SNr] (*Aron, 2011*; *Aron and Poldrack, 2006*; *Schmidt et al., 2013*). Divergence in this pathway is said to be crucial for the broadcast of stop-related activity from the associative/limbic territory of the STN to GABAergic neurons in the skeletomotor and oculomotor territories of GPi and SNr, respectively. The need for divergence at some point in the circuit is reinforced by evidence that reactive stopping behavior is associated with an apparently global suppression of motor and even cognitive function (*Wessel and Aron, 2017*). Limited evidence supports divergence in the STN-GPi/SNr pathway, however (*Hazrati and Parent, 1992*), while multiple lines of evidence indicate instead that this axonal projection is highly specific and topographically organized (*Shink et al., 1996*; *Smith et al., 1998*; *Kolomiets et al., 2001*; *Kelly and Strick, 2004*; *Miyachi et al., 2006*). Potential alternative pathways include the direct STN projection to the tegmental pedunculopontine nucleus

(PPN) (*Shink et al., 1996*; *Hazrati and Parent, 1992*; *Parent and Smith, 1987*; *Nakano et al., 1990*; *Smith et al., 1990*) as well as the disynaptic STN-GPi/SNr-PPN pathway. It is interesting to note in this respect that selective activation of cholinergic neurons of the ventral PPN triggers a short-latency and wide-spread inhibition of motor output (*Takakusaki et al., 1993*; *Takakusaki et al., 2016*; *Kohyama et al., 1998*) in addition to a range of alerting and behavior-modulating functions (*Mena-Segovia et al., 2004*; *Gut and Winn, 2016*). Further study is required to clarify the degree of divergence and specific sites of termination of projections from the ventromedial STN to the PPN and GPi/SNr. And, as mentioned in the Introduction, it is quite likely that reactive stopping such as that studied here depends on a cascade of movement-suppressive processes and multiple brain pathways (*Pouget et al., 2017*; *Schmidt and Berke, 2017*).

Also worth discussing is our frequent observation in the STN of peri-movement increases in activity that were truncated immediately following a stop-signal (*Figure 4A*), activity termed movement-like due to its similarity in form to agonist muscle EMG (*Figure 5A*). Making sense of this type of activity is complicated by the widely-held view that the STN influences motor control centers primarily via direct excitation of the GABAergic output neurons of the BG (in GPi and SNr) (*Nambu et al., 2002*; *Wichmann et al., 1994*; *Hamada and DeLong, 1992*). When transmitted via this pathway, peri-movement increases in STN activity should be movement-suppressive, and the sudden stop-related truncation of activity, movement-facilitator – both of which effects are antithetic to the roles hypothesized for the STN in reactive stopping (*Aron and Poldrack, 2006*). One recently-proposed solution is that another STN efferent pathway, which inhibits BG output neurons via a disynaptic pathway through the external pallidum (GPe), may dominate function of the circuit in certain behavioral contexts (*Jantz et al., 2017*). That, however, does not address the central challenge of explaining the simultaneous presence in the STN of *Stop*-related and movement-related activity patterns. Answers to this puzzle may come from research into the organization of projections from STN subregions, the activity of distinct STN neuronal sub-types during stop-task performance (*Sato et al., 2000*), or how this seemingly-paradoxical combination of STN activities is interpreted in downstream nuclei.

The present results concur in general terms with existing lines of evidence that implicate the ventromedial STN in various clinical disorders of inhibitory control. Case studies have shown that DBS in the ventral STN produces greater effects on inhibitory control behaviors than does stimulation in the dorsal STN (*Greenhouse et al., 2013*; *Hershey et al., 2010*). In fact, this anatomic location is a current target for STN-DBS to treat severe obsessive compulsive disorder (*Mallet et al., 2008*), which is considered a disorder of inhibitory control. And local micro-infusions of bicuculline (a GABA antagonist) only into that part of the STN induced stereotypies and hyperactive behaviors (*Karachi et al., 2009*). The present study is the first to show that single unit encoding of reactive stopping also is restricted to the anterior ventromedial STN. It is important to acknowledge that inhibitory control is a general term that encompasses a disparate array of behavioral faculties and subservient brain circuits. Our results reinforce the view that the ventromedial STN is an important node in one of those brain circuits, but the range of roles played by this circuit in specific behaviors and behavioral disorders remains to be determined. It is interesting to note that activity related to rapid response switching (i.e., *Switch* cells) was also confined to the same anatomic hotspot (*Figure 7C*). Further research is needed to determine whether the *Stop*- and *Switch*-related activities identified here reflect truly distinct processes or rather graded mixtures of signals reflecting different aspects of reactive control.

## Materials and methods

### Animals

Two rhesus monkeys (monkey C, 8 kg, male; and monkey H, 6 kg, female) were used in this study. Procedures were approved by the Institutional Animal Care and Use Committee of the University of Pittsburgh (protocol number: 12111162) and complied with the Public Health Service Policy on the humane care and use of laboratory animals (amended 2002). When animals were not in active use, they were housed in individual primate cages in an air-conditioned room where water was always available. The monkeys' access to food was regulated to increase their motivation to perform the task. Throughout the study, the animals were monitored daily by an animal research technician or

veterinary technician for evidence of disease or injury and body weight was documented weekly. If a body weight <90% of baseline was observed, the food regulation was stopped.

## Apparatus

Monkeys were trained to execute reaching movements with the left arm using a torquable exoskeleton (KINARM, BKIN Technologies, Kingston, Ontario, Canada). This device had hinge joints aligned with the monkey's shoulder and elbow and allowed the animal to make arm movements in the horizontal plane. Motors and encoders attached to the mechanical linkage provided angular position of the joints and applied torques when desired (flexion or extension) to both shoulder and elbow. Visual targets and cursor feedback of hand position were presented in the horizontal plane of hand movements by a virtual-reality system. A detailed description of the apparatus can be found in *Pasquereau and Turner, 2013*.

## Go/NoGo-countermanding task

This task was designed to manipulate different modes of movement suppression (i.e., reactive and proactive controls) and to disentangle possible confounding signals (i.e., stop-related, attention-related or switch-related). We combined the two paradigms used most widely to investigate inhibitory functions and impulse control behaviors: the stop-signal task and the Go/NoGo task. Specifically, in our paradigm, we used a switch-signal (*blue* cue) to unpredictably countermand the initially planned response: going or stopping, depending of the task conditions.

In our Go/NoGo-countermanding task, the monkey was required to align the cursor with a series of distinct visual targets (radius: 1.8 cm) displayed in succession. A trial began when a gray-filled target appeared at a start position proximal to the monkey and the monkey made the appropriate joint movements to align the cursor with this target. The monkey maintained this position for a random-duration hold period (2.9–5.1 s) during which: (a) an instruction cue was displayed at the start position (0.7–1.3 s after hold period start, 0.5 s duration), and (b) a peripheral target was displayed (0.7–1.3 s after offset of the instruction cue, at the same location for all trials, 7.5 cm distal to the start position). Two possible instruction cues were presented in random order across trials with equal probability: a *green* cue indicated a Go trial in which subsequent disappearance of the gray fill from the start position target (the trigger, 0.5–1.5 s after appearance of the peripheral target) cued the animal to promptly move the cursor to the peripheral target (<0.8 s); or alternatively, a *red* cue indicated a NoGo trial in which the animal was required to continue holding the cursor within the start position upon subsequent disappearance of the gray fill from the start target. At the end of each successful trial food reward was delivered via a sipper tube attached to a computer-controlled peristaltic pump (~0.5 ml, puree of fresh fruits and protein biscuits). The temporal delays to reward were calibrated to be equivalent across task conditions. Reward was delivered either 0.7–1.3 s after movement for Go trials or 1.3–1.9 s after withholding movement for NoGo trials.

On 25% of all trials (both Go and NoGo conditions), the trigger was followed after a short delay (the switch-signal delay, SSD) by a switch-signal (*blue* cue, 0.3 s duration). The SSDs varied randomly from trial-to-trial between 200–450 ms. This range of SSDs was selected so that the monkey were able to perform the task with a limited number of errors (5–25% of switch-signal trials). When the switch-signal appeared after a Go cue, the monkey had to inhibit the pending movement and continue to hold the cursor within the start position until reward delivery. In this context, the switch-signal was equivalent to a stop-signal (referred as *switch-stop* trials). If the monkey generated a movement to the peripheral target during switch-stop trials (*stop-failure* trials), no reward was given and a time-out of 4 s occurred to penalize the animal. For the NoGo condition, the monkey was normally required to not respond to the trigger signal, except when the blue switch-signal appeared. Here, the switch-signal was equivalent to a sudden go-signal (*switch-go* trials), thereby cueing the animal to capture the peripheral target to receive a reward. Successive trials were separated by 1.5–2.5 s inter-trial intervals (randomly distributed at 100 ms resolution), during which the screen was black.

## Proprioceptive stimulation

On one-third of the trials (selected at random), single flexing or extending torque impulses (0.34 Nm – 70 ms duration with 10 ms for onset and offset ramps) were applied to the monkey's arm

(shoulder +elbow) by two torque motors at an unpredictable time beginning 200–500 ms after capture of the initial target (i.e., before the instruction period). Each torque impulse induced multi-joint angular displacements causing a sudden stretch of arm extensor or flexor muscles. The animal was required to rapidly counter the perturbation and return its arm to the initial pre-impulse position to continue the current trial.

## Surgery

After reaching asymptotic task performance, animals were prepared surgically for recording using aseptic surgery under Isoflurane inhalation anesthesia. An MRI-compatible plastic chamber (custom-machined PEEK, 28 × 20 mm) was implanted with stereotaxic guidance over a burr hole allowing access to the STN. The chamber was positioned in the parasagittal plane with an anterior-to-posterior angle of 20°. The chamber was fixed to the skull with titanium screws and dental acrylic. A titanium head holder was embedded in the acrylic to allow fixation of the head during recording sessions. For electromyographic (EMG, in monkey C only) recording, pairs of Teflon-insulated multi-stranded stainless steel wires were implanted into multiple arm muscles: biceps longus, triceps lateralis, anterior deltoid and posterior deltoid. Prophylactic antibiotics and analgesics were administered post-surgically.

## Localization of the recording site

The anatomical location of the STN and proper positioning of the recording chamber to access it were estimated from structural MRI scans (Siemens 3 T Allegra Scanner, voxel size of 0.6 mm). An interactive 3D software system (Cicerone) was used to visualize MRI images, define the target location and predict trajectories for microelectrode penetrations (*Miocinovic et al., 2007*). Electrophysiological mapping was performed with penetrations spaced 1 mm apart. The boundaries of brain structures were identified based on standard criteria including relative location, neuronal spike shape, firing pattern, and responsiveness to behavioral events (e.g. movement, reward). By aligning microelectrode mapping results (electrophysiologically characterized X-Y-Z locations) with structural MRI images and high resolution 3-D templates of individual nuclei derived from an atlas (*Martin and Bowden, 1996*), we were able to gauge the accuracy of individual microelectrode penetrations and determine chamber coordinates for the STN.

## Recording and data acquisition

During recording sessions, a glass-coated tungsten microelectrode (impedance: 0.7–1 MOhm measured at 1000 Hz) was advanced into the target nucleus using a hydraulic manipulator (MO-95, Narishige). Neuronal signals were amplified with a gain of 10K, bandpass filtered (0.3–10 kHz) and continuously sampled at 25 KHz (RZ2, Tucker-Davis Technologies, Alachua FL). Individual spikes were sorted using Plexon off-line sorting software (Plexon Inc., Dallas TX). The timing of detected spikes and of relevant task events was sampled digitally at 1 kHz. Horizontal and vertical components of eye position were recorded using an infrared camera system (240 Hz; ETL-200, ISCAN, Woburn, MA). EMG signals were differentially amplified (gain = 10 K), band-pass filtered (200 Hz to 5 kHz), rectified and then low-pass filtered (100 Hz). EMG data were collected during only a subset of data recording sessions (n = 55).

## Analysis of behavioral data

Data were collected after animals had extensive experience with the Go/NoGo-countermanding task (>6 months, with >600 trials per session). Kinematic data derived from the exoskeleton were numerically filtered and combined to obtain the cursor position. The time of response onset, used for determining reaction times (RTs), was determined as the point at which the cursor crossed the outline of the start position target. The Go/NoGo paradigm allowed an animal to decide proactively whether or not to initiate a target capture movement in response to the trigger cue. To test if animals did use such a proactive response strategy, we calculated a behavioral index reflecting the preparation cost by subtracting RTs measured on *go* trials from those on *switch-go* trials.

Two key behavioral measures that characterize performance in the stop-signal task are (1) the inhibition function and (2) the RT distribution for the responses on *go* trials. The inhibition function plots the proportion of *stop-failure* trials (i.e., trials in which the animal fails to stop the incipient

movement) as a function of the SSD. Classically, the probability of erroneous initiation of the movement increases as the SSD increases. The inhibition function has been modeled as a race between a process that initiates movement (Go) and a process that inhibits the response (Stop) (*Logan et al., 1984*). If the Stop process reaches a theoretical threshold before that the Go-related activity is fully processed, the pending movement is suppressed. Otherwise, if the Go process finishes before the Stop process, the movement is generated. As previously described in detail, the race model permits estimation of a stop-signal reaction time (SSRT), which is the time required to inhibit the planned movement. We estimated SSRTs for each session using the quantile method (*Congdon et al., 2012*; *Band et al., 2003*; *Williams et al., 1999*), as follows. First, RTs on correct *go* trials were arranged in ascending order, and the RT corresponding to the same proportion of *stop-failure* trials was selected. For example, for a *P*(failure) of 0.35 we selected the RT 35% up the ordered list. The SSD was then subtracted from this quantile RT, providing an estimate of the time required to stop the response. For each selected SSDs (bins = 28 ms) a value of the SSRT was computed, and then the overall behavioral estimate of cancellation time in a given session was obtained by averaging all the SSRTs. Note that we used six different time bins in the inhibition function in order to optimize the consistency of SSRTs across sessions. All of the data analyses were performed using custom scripts in the MATLAB environment (Mathworks, available online in *Source code 1* File - Custom Matlab code).

## Neuronal data analysis

Neuronal recordings were accepted for analysis based on electrode location, and recording quality and duration (>250 trials). Adequate single unit isolation was verified by testing for appropriate refractory periods (>1 ms) and waveform isolation (signal/noise ratio superior to 3 s.t.d.).

The logic of the stop-signal task sets two criteria a single neuron or a population must meet to play a causally sufficient role in the behavioral response. First, the emitted signal must be different when a movement is initiated versus when it is stopped. Second, this difference must appear within the SSRT; otherwise, it would be too late to drive response inhibition. Following this line of reasoning, we investigated Stop-related activity in our sample of STN neurons by comparing the activity during successful *switch-stop* trials with the activity recorded during the subset of *go* trials in which movement initiation would have been stopped if the switch signal had been presented at the same SSDs. This subset of *go* trials, which we refer to as latency-matched trials, was defined as those with RTs greater than the sum of the SSD and the SSRT calculated from the same data. In the classical approach to this type of analysis, single-unit data are aligned on the time of trigger presentation separately for a series of constant SSDs (*Scangos and Stuphorn, 2010*; *Mirabella et al., 2011*; *Hanes et al., 1998*). In our paradigm, however, SSDs varied randomly from trial-to-trial between 200–450 ms (time resolution = 1 ms), making such an approach impossible in many cases because the same delays between task events were seldom repeated more than twice during any one neuronal recording (probability of occurrence = 1/250). Nevertheless, for the small subset of recordings amendable to such an analysis, we found that application of an adapted version of the 'classic' method yielded results congruent with those from the standard method detailed below (compare the same single-unit data analyzed using 'classical' and standard approaches in *Supplementary file 2* and *Figure 3A*, respectively).

Consequently, to identify Stop-related encodings in neuronal data, rasters and spike density functions were aligned on the time of the Switch-signal. Spike density functions were obtained by convolving spike trains with a Gaussian kernel function (kernel width 20 ms). For *latency-matched go* trials, we used the possible times at which the Switch-signal would have occurred by shuffling delays (SSD: 200–450 ms, 1 ms resolution) 1000 times, and by excluding trials with RT < SSD + SSRT. In order to optimize the time resolution, the STN activity was compared between trials using a sliding window procedure combined with a receiver operating characteristic (ROC) analysis. For each step of 1 ms, we calculated the area under the ROC curve (AUC) derived from the respective spike count distributions detected in a 60 ms test window. The AUC provides a quantitative measure of the separation between two distributions of activity. An AUC value of 0.5 signifies that the two distributions are completely overlapped, whereas an extreme value of 0 or 1 signifies that the two distributions do not overlap. The time of differential activity (i.e., the neural cancellation time) was determined from the evolution of the ROC area over time and was defined as the time at which the AUC differed from a 500 ms control period preceding the Switch-signal (2-tailed *t*-test, $p < 0.01$). If the neural

cancellation time occurred before the SSRT, the STN signal was deemed to qualify as *switch-stop* related activity. To estimate the reliability of this method, we repeated the full procedure 1000 times (neural alignment, AUC measure, and estimation of the neural cancellation time) and calculated the 95% confidence interval of the neural cancellation times yielded for each neuron. We found that the mean temporal resolution for estimating neural cancellation times (i.e., characterized by the size of the confidence interval) was less than 3 ms (see *Supplementary file 2*).

To disentangle confounding signals such as those related to attentional detection of the cue or those involved in behavioral switching function, we compared STN responses observed on successful *switch-stop* trials with the activity recorded on successful *switch-go* trials. The same sliding window ROC method was used to test whether respective activities were significantly different between trials. Briefly, we tested whether the AUC differed from a 500 ms control period preceding the Switch-signal (2-tailed *t*-test, p<0.01). A neuron was judged to be specifically involved in stopping if it generated a countermanding-related signal on *switch-stop* trials that exceed the neuronal response emitted on *switch-go* trials. Otherwise, if a neuron generated an equivalent or greater response on *switch-go* trials, then it was considered to be related to attentional or switching function. We referred to cells with this latter form of non-specific neuronal activity as Switch cells.

We also tested STN neurons for their responsiveness to proprioceptive input from the proximal arm and thus their membership in the STN skeletomotor circuit and potential involvement in the short latency activation of antagonist muscles related to stopping. We aligned the STN activity on the time of sudden torque motor-induced rotations of the animal's elbow and shoulder joints. Mean peri-torque rasters and SDFs were constructed separately for two torque directions (flexion and extension). A phasic response to a torque perturbation was detected by comparing spike counts in a 60 ms test window to spike counts time-shuffled (ROC procedure with a sliding step of 1 ms). The threshold for significance was estimated from the AUC values calculated within a 500 ms control period preceding the torque (2-tailed *t*-test, p<0.01). A neuron was judged to be torque-related if it generated a significant short-latency response (<200 ms) for at least one movement direction.

Following the same procedures as described above, we also tested STN single-unit activity for responsiveness to an error signal or reward delivery when the animal, respectively, failed or succeeded to stop during switch-stop trials. Mean peri-event rasters and SDFs were constructed around the trial outcomes (error or reward) and phasic responses were detected by comparing spike counts in a 60 ms test window to spike counts time-shuffled (2-tailed *t*-test, p<0.01). A neuron was judged to be error- or reward-related if it generated a significant response within 500 ms of the error or reward signal.

To further characterize individual STN neurons, the Go/NoGo-countermanding task was also devised to study the proactive control of movement suppression. Following the well-defined logic of the Go/NoGo task, if a neuron's activity is related to proactive movement suppression, the activity preceding the trigger on *no-go* trials must exceed the activity on *go* trials. Alternatively, for a neuron's activity to be related to movement preparation, the activity preceding the response on *go* trials must exceed that on *no-go* trials. To this end, we compared STN activity between *go* and *no-go* trials using the same sliding window procedure and ROC method as above. The neuronal activity was aligned on the time of the instruction and on the time of the response trigger, and we calculated the AUC derived from the respective spike count distributions. The AUC values compared spike counts from the instruction-trigger interval against a cell's 500 ms baseline preceding the instruction period (2-tailed *t*-test, p<0.01).

Although monkeys were not required to control their gaze while performing the task, we conducted a multiple regression analysis to estimate how task-related neuronal activity around the time of the switch-signal was related to eye positions and saccades. We tested for relationships between neuronal activity and continuous signals reflecting eye position ($b_1$ and $b_2$ corresponded to the horizontal and vertical components) and velocity ($b_3$ and $b_4$). For each neuron, we counted spikes in a 50 ms test window stepped in 1 ms increments from $-0.5$ to $+1$ s relative to the switch signal. Specifically, we used the model:

$$f(t) = b_0 + \begin{bmatrix} b_1 \\ b_2 \end{bmatrix} x(t) + \begin{bmatrix} b_3 \\ b_4 \end{bmatrix} \dot{x}(t) + \varepsilon \qquad (1)$$

where $f$ is the spike count at time t, $b_0$ is the baseline firing rate, x is eye position, $\dot{x}$ is velocity, and ε

depicts the residual error. To test whether an $R^2$ value was significant, we shuffled the inter-spike intervals repeatedly (1000 times) and compared the actual $R^2$ value to the 99% confidence interval of $R^2$ values yielded by shuffling.

### Analysis of EMG data

Electromyographic signals from proximal arm muscles were processed in a manner that paralleled our analysis of neuronal signals using the mean filtered activity across trials for each recording session. By comparing the activity during successful *switch-stop* trials with the activity recorded during *latency-matched go* trials, we determined the time of differential activity for each muscle. Here again we used the evolution of the AUC values across time to detect changes related to the switch-signal (2-tailed *t*-test, p<0.01).

## Acknowledgements

This work was supported by National Institute of Neurological Disorders and Stroke at the National Institutes of Health (grant numbers R01NS091853 and R01NS070865 to RST) and the Center for Neuroscience Research in Non-human primates (CNRN, 1P30NS076405-01A1).

## Additional information

### Funding

| Funder | Grant reference number | Author |
| --- | --- | --- |
| National Institute of Neurological Disorders and Stroke | NS091853 | Robert S Turner |
| National Institute of Neurological Disorders and Stroke | NS070865 | Robert S Turner |
| National Institute of Neurological Disorders and Stroke | NS076405-01A1 | Robert S Turner |

The funders had no role in study design, data collection and interpretation, or the decision to submit the work for publication.

### Author contributions

Benjamin Pasquereau, Conceptualization, Data curation, Software, Formal analysis, Validation, Investigation, Visualization, Methodology, Writing—original draft, Writing—review and editing; Robert S Turner, Conceptualization, Resources, Formal analysis, Supervision, Funding acquisition, Validation, Investigation, Visualization, Methodology, Project administration, Writing—review and editing

### Author ORCIDs

Benjamin Pasquereau https://orcid.org/0000-0003-2855-0672
Robert S Turner http://orcid.org/0000-0002-6074-4365

### Ethics

Animal experimentation: This study was performed in strict accordance with the recommendations in the Guide for the Care and Use of Laboratory Animals of the National Institutes of Health. All of the animals were handled according to an approved institutional animal care and use committee (IACUC) protocol (#15106967) of the University of Pittsburgh. All surgery was performed under Isoflurane anesthesia, and every effort was made to minimize pain and suffering.

### Decision letter and Author response

Decision letter https://doi.org/10.7554/eLife.31627.018
Author response https://doi.org/10.7554/eLife.31627.019

## Additional files

### Supplementary files

• Supplementary file 1. Table showing numbers of STN neurons found in each category per monkey.
DOI: https://doi.org/10.7554/eLife.31627.014

• Supplementary file 2. Validation of the method used to detect the neural cancelation time. (A-B) Population-averaged activities of (A) Go cells and (B) NoGo cells that showed a short-latency response (<300 ms) to the Switch-go signal. Spike density functions aligned on switch-go signal presentation (*green*) and the equivalent time in no-go trials (*gray*) were normalized by subtracting the baseline activity (500 ms before the signal) and grouped according to the response pattern evoked in neuronal activity: increase (*left*) or decrease (*right*) in firing relative to no-go trials. The vertical width of the spike density function line indicates the population SEM. To compare the firing rate between trials, we used the same ROC analysis (p<0.01). The arrows in AUC values indicate the times of first differential activity. (C) Distribution of peak AUC values for the Go/NoGo comparison plotted versus the AUC values for the switch-go comparison. No relationship was observed in the STN between both reactive and proactive encodings ($\chi^2$=1.14, p=0.29).
DOI: https://doi.org/10.7554/eLife.31627.015

• Source code 1. Custom Matlab code for analysis.
DOI: https://doi.org/10.7554/eLife.31627.016

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
