## [Decision Letter]

Thank you for submitting your article "A selective role for ventromedial subthalamic nucleus in inhibitory control" for consideration by *eLife*. Your article has been favorably evaluated by Timothy Behrens (Senior Editor) and three reviewers, one of whom, Naoshige Uchida (Reviewer #1), is a member of our Board of Reviewing Editors. The following individual involved in review of your submission has agreed to reveal their identity: Veit Stuphorn (Reviewer #3).

The reviewers have discussed the reviews with one another and the Reviewing Editor has drafted this decision to help you prepare a revised submission.

Summary:

The subthalamic nucleus (STN) is thought to play a critical role in switching behaviors in response to a cue (e.g. a stop signal). Pasquereau and Turner conducted a very careful characterization of STN neuron firing in monkeys performing a Go/No-go countermanding arm movement task (a combination of a stop signal task and a Go/No-go task). The authors aimed to dissociate various confounding factors. First, because stop signals were presented infrequently, these signals would trigger strong attentional responses. Therefore, the authors examined the effect of go signals that indicated a switch from a No-go to a Go response (instead of a stop signal indicating a switch from a Go to a No-go response). The presence of these types of trials allowed the authors to test whether a neural response is related to attentional modulation since both the go and stop signals are equally infrequent. By comparing neural activities responding to a cue indicating a No-Go response versus those responding to a stop signal in cued Go trials, the authors were able to examine whether a stop signal response was related to reactive stopping or proactive action inhibition. In addition, the use of both go and stop signals allowed the authors to examine whether a switch activity is specific to a switch to a go or a switch to a stop, or both. The authors also examined whether the neural activity was correlated with mere motor parameters or proprioceptive responses. Through these examinations, the authors found neurons encoding a reactive stop or switching to go, and these neurons were localized to the ventromedial 'limbic' region of STN instead of the 'skeltomotor' regions of the STN.

All the reviewers thought that the study is excellent, and timely. A role of the STN in response inhibition has been proposed for a long time, but this was mainly based on human fMRI data and general notions about the function of different pathways in the basal ganglia. Recently, the role of STN has been investigated in rodents, but there were a number of technical differences in the design of the experiment relative to the previous experiments in humans and non-human primates. Thus, it is important that this issue is investigated in monkeys using single unit recording, whose high temporal and spatial resolution allows the disambiguation of different potential cognitive signals and of the functional role of different cortico-basal ganglia circuits. In addition, the 'classic' stop signal paradigm potentially conflates a number of different cognitive processes. This problem has been elegantly addressed in the new innovative experimental design that combines aspects of the Go/NoGo and the Stop Signal tasks. The findings are clear and interesting. The STN contains neurons with an activity profile that is sufficient to be causally involved in controlling response initiation and suppression. This finding confirms the prevailing theories. An unexpected aspect of the findings is that these neurons were not found in the 'skeletomotor' regions of the STN, but instead in the ventromedial STN, which receives input from limbic structures.

In summary, all the reviewers thought that this work will likely become an important (one reviewer said a landmark) study in the field of inhibitory control. While the reviewers are enthusiastic, there are a few questions and suggestions that may improve the manuscript.

Essential revisions:

1) A somewhat surprising finding is that the stop-related STN neurons were *not* in the motor sections of the STN. At a glance, this appears to contradict with the author's emphasis on a causal involvement of these neurons in switching responses. The reviewers noted that there are at least two possibilities that these neurons may affect motor responses. First, such stop activity may have influence on motor execution through overlapping cortico-basal ganglia pathways (the segregation of parallel pathways might not be complete). Alternatively, such limbic STN activity may have influence on limbic cortical areas that in turn, affect motor execution indirectly. Please discuss possible route through which the limbic STN regulates motor outputs.

2) Although the present results provide novel insights, there are some previous studies that provided overlapping observations. Please discuss more explicitly differences from the previous recording studies (particularly, Schmidt et al., 2013 and Isoda and Hikosaka, 2008). It is very helpful to discuss what findings are invariant to detailed task parameters and species, and what findings differ among these studies.

3) Please clarify the way the comparison between the go and the switch-stop trials is done. In the go trials there is no switch signal. In the 'classic' countermanding studies (eye: Hanes et al., 1998; arm: Scangos and Stuphorn, 2010; Mirabella et al., 2011) the trials were therefore aligned on target onset and neuronal activity across multiple SSDs was compared. In the current manuscript, the authors use a shuffling (bootstrapping?) technique, where they draw 1000 trials randomly from the go trial distribution, pick a SSD in proportion to the actually used times, throw the trial out, if RT is faster than SSD, and add the activity to the 'switch-aligned' average, if RT is slower than SSD. (Please make the description clearer, if there is a misunderstanding.) This procedure appears to be fine. However, there remains a few concerns. First, we would like to see the 'classic' alignment on target onset (as a supplementary figure for a neuron where enough trials exist across at least two SSDs). This would help convince that this new procedure works accurately. Second, we were confused about the trial number in Figure 3. How many trials are shown in the raster plots for the two types of trials? There are certainly not 1000 trials shown in the case of the 'switch-stop' trials. So, how were the trials picked? Also, should the procedure not be to assign randomly SSDs to all go trials, get an AUC measure and then repeat this 1000 times to get an estimate of the reliability of the AUC strength and the time of significant differences (if any).

---

## [Author Response]

Essential revisions:1) A somewhat surprising finding is that the stop-related STN neurons were not in the motor sections of the STN. At a glance, this appears to contradict with the author's emphasis on a causal involvement of these neurons in switching responses. The reviewers noted that there are at least two possibilities that these neurons may affect motor responses. First, such stop activity may have influence on motor execution through overlapping cortico-basal ganglia pathways (the segregation of parallel pathways might not be complete). Alternatively, such limbic STN activity may have influence on limbic cortical areas that in turn, affect motor execution indirectly. Please discuss possible route through which the limbic STN regulates motor outputs.

We added a paragraph to the Discussion that addresses this question directly (ninth and tenth paragraphs).

2) Although the present results provide novel insights, there are some previous studies that provided overlapping observations. Please discuss more explicitly differences from the previous recording studies (particularly, Schmidt et al., 2013 and Isoda and Hikosaka, 2008). It is very helpful to discuss what findings are invariant to detailed task parameters and species, and what findings differ among these studies.

We revised the Discussion to consolidate most comparisons with previous similar studies into one new paragraph (sixth paragraph).

3) Please clarify the way the comparison between the go and the switch-stop trials is done. In the go trials there is no switch signal. In the 'classic' countermanding studies (eye: Hanes et al., 1998; arm: Scangos and Stuphorn, 2010; Mirabella et al., 2011) the trials were therefore aligned on target onset and neuronal activity across multiple SSDs was compared. In the current manuscript, the authors use a shuffling (bootstrapping?) technique, where they draw 1000 trials randomly from the go trial distribution, pick a SSD in proportion to the actually used times, throw the trial out, if RT is faster than SSD, and add the activity to the 'switch-aligned' average, if RT is slower than SSD. (Please make the description clearer, if there is a misunderstanding.) This procedure appears to be fine. However, there remains a few concerns. First, we would like to see the 'classic' alignment on target onset (as a supplementary figure for a neuron where enough trials exist across at least two SSDs). This would help convince that this new procedure works accurately. Second, we were confused about the trial number in Figure 3. How many trials are shown in the raster plots for the two types of trials? There are certainly not 1000 trials shown in the case of the 'switch-stop' trials. So, how were the trials picked? Also, should the procedure not be to assign randomly SSDs to all go trials, get an AUC measure and then repeat this 1000 times to get an estimate of the reliability of the AUC strength and the time of significant differences (if any).

To address this point we modified the text (in the Materials and methods section entitled “Neuronal data analysis” second paragraph) and we created a new Supplementary file 2.

To investigate Stop-related activity in our sample of STN neurons, we compared the activity during successful switch-stop trials with the activity recorded during the subset of go trials in which movement initiation would have been stopped if the switch signal had been presented at the same SSDs (RT < SSD + SSRT). As described by the reviewers, the classical approach is to align single-unit data on the time of trigger presentation separately for a series of constant SSDs (Hanes et al., 1998; Scangos and Stuphorn, 2010; Mirabella et al., 2011). In our paradigm, however, SSDs varied randomly from trial-to-trial between 200–450 ms (time resolution = 1 ms), making such an approach impossible in many cases because the same delays between task events were seldom repeated more than twice during any one neuronal recording (probability of occurrence = 1/250).

Nevertheless, for the small subset of recordings amendable to such an analysis, we found that application of an adapted version of the ‘classic’ method yielded results congruent with those from our standard method. Example single-unit data analyzed using “classical” and standard approaches are shown in Supplementary file 2 and Figure 3, respectively.

To identify Stop-related encodings in neuronal data, rasters and spike density functions were aligned on the time of the Switch-signal. For latency-matched go trials, we used the possible times at which the Switch-signal would have occurred by shuffling delays (SSD: 200–450-ms, 1-ms resolution) 1,000 times, and by excluding trials with RT < SSD + SSRT. To estimate the reliability of our method, we repeated the full procedure 1,000 times (neural alignment, AUC measure, and estimation of the neural cancellation time) and calculated the 95% confidence interval of the neural cancellation times yielded for each neuron. We found that the mean temporal resolution for estimating neural cancellation times (i.e., characterized by the size of the confidence interval) was less than 3-ms (see Supplementary file 2 in which we show the distribution of confidence intervals for estimation of cancellation times), making our method quite accurate.

In Figure 3, only a subset of trials (randomly chosen) is shown in the raster of go trials to aid visualization of spike occurrences. The density of points would impair visualization if the full 1000 trials had been included in the figure. We have added a note in the figure legend to explain this.